# Static and Dynamic Optical Analysis of Micro Wrinkle Formation on a Liquid Surface

**DOI:** 10.3390/mi12121583

**Published:** 2021-12-19

**Authors:** Antariksh Saxena, Costas Tsakonas, David Chappell, Chi Shing Cheung, Andrew Michael John Edwards, Haida Liang, Ian Charles Sage, Carl Vernon Brown

**Affiliations:** 1SOFT Group, School of Science and Technology, Nottingham Trent University, Clifton Lane, Nottingham NG11 8NS, UK; antarikshsaxena@gmail.com (A.S.); costas.tsakonas@ntu.ac.uk (C.T.); david.chappell@ntu.ac.uk (D.C.); A.Edwards@lboro.ac.uk (A.M.J.E.); ian.sage@ntu.ac.uk (I.C.S.); 2Imaging & Sensing for Archaeology, Art History & Conservation (ISAAC) Group, School of Science and Technology, Nottingham Trent University, Clifton Lane, Nottingham NG11 8NS, UK; sammy.cheung@ntu.ac.uk (C.S.C.); haida.liang@ntu.ac.uk (H.L.)

**Keywords:** dielectrophoresis, dielectric material, optofluidics, optical diffraction, optical coherence tomography

## Abstract

A spatially periodic voltage was used to create a dielectrophoresis induced periodic micro wrinkle deformation on the surface of a liquid film. Optical Coherence Tomography provided the equilibrium wrinkle profile at submicron accuracy. The dynamic wrinkle amplitude was derived from optical diffraction analysis during sub-millisecond wrinkle formation and decay, after abruptly increasing or reducing the voltage, respectively. The decay time constant closely followed the film thickness dependence expected for surface tension driven viscous levelling. Modelling of the system using numerical solution of the Stokes flow equations with electrostatic forcing predicted that wrinkle formation was faster than decay, in accord with observations.

## 1. Introduction

The ability to create a uniformly flat film of a viscous liquid on a solid surface is important in a wide range of coating, painting, and varnishing applications [1,2], including microelectronics manufacturing [3,4], architecture and decoration [5], and protection and art conservation [6,7]. Such films are susceptible to distortions of the liquid-air interface occurring during initial shaping, for example as brush strokes while painting, or after deposition, for example due to substrate roughness [8] or due to an impinging air flow over the film [9]. In wet paint systems, distortions in the liquid surface during solvent evaporation and curing can ruin the quality of the final finish [10] and in spin-coated films for microelectronics non-uniformity leads to reduced device yield [4].

A low amplitude distortion at the air interface of a non-evaporating homogeneous liquid film will decrease in amplitude over time under the combined effects of surface tension, *γ*, and gravity, *g*, while being resisted by the viscosity, *η*, of the liquid film [11]. The physical mechanism driving the decay depends on the ratio of the wrinkle pitch, *p*, to the capillary length, *L*c = (γ/Δ*ρg*)^1/2^, where Δ*ρ* is the difference in density between the liquid and air phases [12]. Distortions in which *L*c > *p* are dominated by surface tension driven levelling and distortions for which *L*c < *p* are primarily levelled by gravitational effects [11].

Conversely, the ability to purposefully create and control the formation of small-scale periodic wrinkles on the surface of liquid films can also be advantageous in applications. The formation of wrinkles on liquid coatings during curing has been used to promote adhesion of an elastomer onto a solid surface [13] and the instability created by the action of electrostatic pressure on a melted polymer-air interface has been used to create periodic patterns on a solid polymer film [14,15]. In optical devices, an electrically programmable periodic wrinkle on an oil film created by dielectrophoresis forces has been used to demonstrate a tuneable diffraction grating [16] and an optofluidic light guide for solar lighting control [17]. The detailed control of the shape of the periodic distortion on the interface between two liquid layers has been demonstrated for potential beam-steering for Fresnel lens based optofluidic applications [18].

Previously, we have demonstrated how dielectrophoresis forces can spread an electrically insulating liquid to a film on a solid surface and, furthermore, create a static programmable wrinkle at the liquid-air interface [16]. A spatially periodic voltage applied via an interdigital array of striped coplanar electrodes embedded into the solid surface produces a surface localized periodic and non-uniform electric field that decays into the underside of the liquid. This causes a dielectrophoresis force to act on the liquid, which is a polarizable dielectric material, in the direction of the increasing magnitude of the electric field [19,20]. This force results in the liquid spreading over the surface, eventually forming into a uniform thin film with average thickness *h*_ave_. Since the magnitude of the electric field is at a maximum above the gaps between the electrode fingers, this also results in the liquid preferentially collecting within these regions and so the liquid-air interface develops a periodic “wrinkle” deformation, i.e., a corrugation, with pitch determined by the underlying electrode structure [16].

Wrinkling is important for a number of applications, as described above, and hence being able to measure wrinkle patterns quantitatively is desirable. Using wrinkling of the surface of a liquid film created by dielectrophoresis forces allows us to vary the static amplitude of the wrinkles straightforwardly by changing the voltage applied to the same liquid film. Our ability dynamically both to form the wrinkles, and then also to allow the wrinkle amplitude to decay after the voltage is significantly reduced (we do not completely remove the voltage to avoid film dewetting), provides a well-defined model system for the study of viscous liquid levelling. Furthermore, we can readily change the volume and the thickness of the liquid film on which the wrinkles are formed by removing or adding liquid, since the film is formed over a constrained area determined by the electrode geometry.

In this work, we demonstrate and develop optical methods to study the static equilibrium profile and the dynamic formation and decay of a periodic wrinkle deformation on the liquid-air interface of a thin film of an electrically insulating liquid. The film and the wrinkle are formed by liquid dielectrophoresis forces, and the pitch *p* is larger than the film thickness whilst also being considerably shorter than the capillary length for the liquid, *h*_ave_
*< p < L*_C_. We demonstrate that Optical Coherence Tomography (OCT) imaging can be used to make precise measurements of a static wrinkle deformation profile on the liquid surface in equilibrium, with sub-micrometer axial resolution.

OCT is a 3D imaging technique based on the Michelson interferometer where the depth profile is derived from the interference pattern generated from the pathlength difference between the sample and the reference path [21]. It is a well-established technique for imaging the surface and subsurface microstructures in biomedical fields, particularly ophthalmology [22,23]. OCT imaging techniques have also been increasingly developed and applied across a broad a range of different disciplines, for example, the non-invasive analysis of paintings for art history and conservation [24], and profiling surface membranes for reconfigurable optics applications [25]. OCT developments and applications which involve imaging liquids and liquid surfaces have included micro-rheology studies of simple and complex fluids [26], analysing capillary waves in biological fluids [27,28], droplet imaging [29] and interface imaging to study surface wettability [30], gelation [31], and evaporation [32].

Informed by the knowledge of the static equilibrium surface profiles obtained from the OCT study, we have employed time resolved optical diffraction measurements to study the sub-millisecond dynamics of the dielectrophoresis driven wrinkle formation and the dynamics of fast capillary driven decay. This technique enables the facile measurement of the wrinkle formation and decay timescales on a selected and localized region of the overall liquid film surface area. We compare our dynamic formation results to new simulations based on approaches reported by Chappell and O’Dea [33], and our dynamic decay results to the analysis of Orchard [11] for a range of liquid film thicknesses.

## 2. Materials and Methods

The device substrate was a borosilicate glass slide, which was coated with indium tin oxide of thickness 25 nm and resistivity 100 Ω/square (Praezisions Glass and Optick GmbH, Iserlohn, Germany). The electrode geometry was etched into the transparent conducting oxide layer using standard photolithography techniques. The electrodes consisted of interdigitated coplanar stripes in the *y*-direction, as shown in Figure 1, covering a square area on the substrate of size 5 mm × 5 mm in the *x*-*y* plane. The electrode linewidths ($w_{e}$) and the size of the gap between electrodes ($w_{g}$) were both equal, $w_{e}=w_{g}=$ 80 µm. The substrate was further coated with a 0.5 μm thick protective solid electrically insulating layer of SU-8 photoresist (Microchem Corp., Westborough, MA, USA).

Different volumes of the dielectric liquid TMP-TG-E (trimethylolpropane triglycidyl ether, CAS Number 3454-29-3, Sigma Aldrich, Darmstadt, Germany) were dispensed onto the centre of the electrode area using a Gilsen (0.1 to 2 μL) pipette. TMP-TG-E was chosen because this liquid combines the favourable properties of a high dielectric constant of $\varepsilon_{r}$ = 13.8 combined with low electrical conductivity and low dielectric loss at the range of frequencies of the A.C. voltages that we used in our experiments (1 kHz to 2.5 kHz). TMP-TG-E has viscosity of η = 0.1867 ± 0.001 N s m^−2^, a surface tension of γ = 0.043 ± 0.001 N m^−1^, a refractive index of $n_{TMP-TG-E}=$ 1.477 at a wavelength of 532 nm, and a mass density of 1157 ± 10 kg m^−3^ [34]. With no applied voltage, TMP-TG-E formed a spherical cap shaped droplet on the SU-8 coated substrates with an equilibrium contact angle of 30 ± 2°. All experiments were performed on the open bench in a temperature-controlled laboratory (21 ± 1 °C).

An A.C. sinewave voltage $V_{R.M.S.}$ was applied to alternate electrodes in the array whilst the interposed electrodes were held at earth potential, as shown in Figure 1. The voltage waveform up to 300 V (R.M.S. voltage values given in text) was created by a waveform generator (TGA1244, Thurlby Thander Instruments Limited, Cambridge, UK) combined with a voltage Amplifier (PZD700, Trek Inc., Medina, New York, NY, USA). Application of a constant voltage $V_{R.M.S.}$ of 50 V to the electrodes was sufficient to cause the dielectric liquid to spread to form a liquid film covering the electrode area. The applied voltage resulted in the formation of a clear “wrinkle” deformation of pitch *p* = 160 μm at the liquid-air surface. This is a consequence of dielectrophoresis forces collecting the liquid in the gap regions between the electrodes. The pitch p is hence determined solely by the electrode linewidth and the size of the gap between electrodes by the relationship $p=w_{e}+w_{g}$. Calculation of the capillary length gives *L*_C_ ≈ 1950 μm >> *p*, indicating that when we subsequently significantly reduce the voltage, the levelling of these induced surface wrinkle distortions will be dominated by capillary forces. Dielectrophoresis forces are independent of polarity and so, in theory, either D.C. or A.C. voltages can be used create the wrinkle deformation. However, in practice A.C. voltages are used to the avoid the D.C. and low frequency shielding effects of the field induced migration of free charges that are inevitably present even in the low electrically conductivity liquid TPM-TG-E. We used voltages of 1 kHz (Section 3.1) and 2.5 kHz (Section 3.2) which provides a suitable compromise that avoids low frequencies that would have caused charge migration and would also have significantly modulated the wrinkle amplitude during each half period, and also avoids high frequencies where the finite slew rate of the amplifier and the finite conductivity of the indium tin oxide electrodes would have caused signal losses.

The static equilibrium surface profile of the liquid film *h*(*x*), as defined in the schematic diagram in Figure 1, was measured using Optical Coherence Tomography (OCT) whilst a constant A.C. voltage was applied to the electrodes. The ultra-high resolution Fourier domain OCT, used at a central wavelength of 810 nm, is described in detail by Cheung et al. [24], and a schematic diagram of the apparatus is shown in Figure 2a. OCT gives cross-section images and is sensitive to changes in refractive indices across interfaces within the cross-section. Unlike confocal microscopy where the axial and the transverse resolution are inter-related by the numerical aperture and focal length of the objective used, in OCT the axial resolution is dependent on the coherence length of the broadband laser source used and it is decoupled from the transverse resolution. This allows OCT to have a wider field of view while still maintaining a high axial resolution. The OCT employed here has an axial resolution of 1.2 µm and a transverse resolution of 7 µm. The OCT system operated with a frame rate of 25 Hz for a 500-line scan of 5 mm in length. The incident optical power on the sample was around 1 mW. Figure 2b shows a raw “BScan” image produced by OCT imaging of the TMP-TG-E film with a constant static A.C. voltage $V_{R.M.S.}$ = 250 V (1 kHz) applied to the electrodes of the device. The surface profile of the film was obtained through Fast Fourier Transform signal processing of the raw interference data captured by the OCT.

The dynamic peak to trough amplitude of the surface wrinkle deformation A(t), as defined in the schematic diagram in Figure 1, was derived from time resolved optical diffraction measurements. Expanded and collimated light from a 10 mW, unpolarised laser diode provided illumination at wavelength λ = 532 nm. The laser light was transmitted through the transparent substrate and the liquid film and diffracted by the periodic phase retardation due to the evolving or decaying periodic surface wrinkle. The time-dependent intensities of the 0, +1 and +2 diffracted orders were recorded using silicon photodetectors (DET36A, Thorlabs, Ely, UK).

## 3. Results

### 3.1. Static Equilibrium Profile of the Liquid Surface Wrinkle

We first demonstrate the use of Optical Coherence Tomography imaging to measure the surface profile of a static periodic surface wrinkle deformation on the spread liquid film. We created and maintained the liquid film and the wrinkle using dielectrophoresis forces. We deposited a 1.48 ± 0.05 μL volume of the liquid TMP-TG-E on to our device substrate and applied an A.C. voltage $V_{R.M.S.}$ = 200 V (1 kHz) to the electrodes. The liquid rapidly spread, within 1 s of the voltage being applied, into a thin film and developed a static periodic surface wrinkle deformation, with the film surface attaining a static equilibrium shape under the influence of the constant applied voltage. We analysed the OCT cross-section BScan image of the liquid film. Firstly, we corrected the curvature of the field as shown on the right hand side of the example BScan profile shown Figure 2b. We then fitted the depth profile (AScan) for the peaks corresponding to the air-liquid and air-substrate interfaces at each *x* position to subpixel accuracy. After accounting for the refractive index of the liquid TMP-TG-E this gave the film thickness at each *x* position. Repeating this procedure for a range of *x* positions yielded the equilibrium surface height profile of the film shown in the labelled plots on the graph of *h*(*x*) versus distance *x* in Figure 3. This height profile represents a cross-section through the centre of the film in the *x*-direction. The 1.48 ± 0.05 μL volume of TMP-TG-E resulted in a spread liquid film having average thickness $h_{ave}$ in the range 57 μm to 60 μm across a significant proportion (*x* = 0.5 mm to *x* = 4.5 mm) of the span of the 5 mm wide region covered by the electrode in the *x*-direction, shown in the upper labelled plot in Figure 3. The device substrate was then cleaned using propan-2-ol before a smaller volume of TMP-TG-E was dispensed onto the electrode area, 0.85 ± 0.05 μL and subjected to the same applied A.C. voltage $V_{R.M.S.}$ = 200 V (1 kHz). The equilibrium surface profile for this film, also extracted from OCT measurements, is shown as the lower labelled plot in the *h*(*x*) versus *x* graph in Figure 3. This lower volume resulted in a spread liquid film with an average thickness $h_{ave}$ in the range 30 μm to 34 μm in the region between *x* = 0.5 mm and *x* = 4.5 mm. It is clear from Figure 3 that the static equilibrium peak to trough amplitude of the wrinkles, A=A(t→∞), is larger on the liquid film that has the lower average thickness that was produced from the lower dispensed volume.

The liquid surfaces were then examined in finer detail (Figure 4). Since OCT imaging depends on the amount of scattered light from a liquid surface, regions with steep features will scatter less light back and therefore be more challenging to image. To overcome this problem, it is necessary to perform averaging over scans, particularly for the higher wrinkle amplitudes shown in Figure 4. Noise on the profiles shown was reduced by averaging over scans on 3 distinct neighbouring regions of the wrinkle after removal of the slow *x*-variation of the average height baseline. The different vertical profiles in Figure 4 were measured with a range of A.C. voltage values applied to the electrodes under the liquid,$\;V_{R.M.S.}$ = 100, 150, 200, 250, 300 V (1 kHz). The vertical positions of the profiles shown are offset for clarity to facilitate shape and amplitude comparisons between them. Figure 4 shows that the liquid-air interface profiles have a peak to trough wrinkle amplitude A that has a monotonic dependence on the applied voltage, with higher amplitude at higher voltage. The plots also show that for a given applied voltage the wrinkle amplitude is higher for the liquid film of lower thickness shown in Figure 4b than for the thicker liquid film in Figure 4a. For example, with an applied voltage of $V_{R.M.S.}$ = 200 V the peak to trough wrinkle amplitude is A = 2.25 ± 0.05 μm on the film with thickness 30 μm to 34 μm in Figure 4b, compared with A = 0.93 ± 0.05 μm on the film with thickness 57 μm to 60 μm in Figure 4a. This effect is a consequence of the fact that the electric fields above the electrodes decay exponentially with distance in the *z* direction. Hence for a thinner liquid film the air-liquid interface is closer to the electrodes compared to a thicker film, and so the former experiences a substantially stronger dielectrophoresis force to create the larger wrinkle deformation. The shapes of the periodic surface wrinkles show greater deviation from an idealized sinusoidal shape for the higher voltage values, particularly for the thinner liquid film.

In previous work in the literature, the shapes and amplitudes of dielectrophoresis induced liquid surface wrinkles, including derived from direct imaging and Mach–Zehnder based interferometric measurements, were analysed using a number of different theoretical approaches [16,20,33,35,36]. An approximate analytical expression has previously been derived, Equation (8) in reference [37], that relates the peak to trough wrinkle amplitude, A, to the square of the voltage, $V_{o}^{2}$ (where $V_{o}=½V_{R.M.S.}$), via a coefficient that depends on the variables $\varepsilon_{r}$, γ, p, and *h*_ave_ that have been defined in the Experimental section and in Figure 1 of this paper. This analytical expression was derived by assuming a spatially periodic potential at *z* = 0 that is a sinusoidal function of *x* and that $h\left(x\right)=h_{ave}+\left(A/2\right)\cos\left(kx\right)$, with the condition that 2πA/p≪1. This expression predicts that the voltage scaled wrinkle peak to trough amplitude will be given by $A/V_{o}^{2}$ = 0.86 × 10^−10^ mV^−2^ for the data shown in Figure 4a, for which the experimentally measured amplitudes are in the range (0.91–1.06) × 10^−10^ mV^−2^, and by $A/V_{o}^{2}$ = 1.87 × 10^−10^ mV^−2^ for the data shown in Figure 4b, for which the experimentally measured amplitudes are in the range (1.85–2.17) × 10^−10^ mV^−2^. Whilst this analytical expression should not be expected to provide precise quantitative predictions, due to the simplified model of the non-uniform electric fields in the system, the sinusoidal profile approximation, and due to the presence of a thin solid dielectric protective film over-coating the electrodes, we do find very good correspondence between the predicted voltage-scaled wrinkle amplitude values and the values measured from our OCT results. The results from OCT imaging shown in Figure 4 illustrate how OCT techniques can be used to obtain high axial resolution profiles of micrometer scale features on a liquid-air surface, across a range of liquid-air wrinkle amplitudes spanning from A = 0.24 μm (100 V, Ω = 1.48 ± 0.05 μL) up to A = 4.42 μm (300 V, Ω = 0.85 ± 0.05 μL).

### 3.2. Dynamic Growth and Decay of the Liquid Surface Wrinkle

We next demonstrate how dynamic optical diffraction techniques can be employed to quantify the time dependent amplitude A(t) of a dielectrophoresis induced periodic wrinkle on a liquid surface during its growth towards equilibrium immediately after a voltage is abruptly applied, and also during its decay after a voltage is abruptly reduced. For a liquid with a relatively low viscosity, both wrinkle growth and wrinkle decay will take place on timescales well below 1 millisecond [16]. We demonstrate here a facile indirect optical diffraction-based measurement technique that exploits the fact that a periodic wrinkle deformation on a liquid-air surface acts as a diffraction grating. If the wrinkle is approximated by a sinusoidal functional dependence, $h\left(x\right)=h_{ave}+\left(A/2\right)\cos\left(kx\right)$, then the spatially periodic optical path variation, ϕ(x), will be given by Equation (1).
(1)$$\phi (x)=\left(\frac{2\pi }{\lambda_{air}}\right)\left[h_{ave}n_{liquid}+\left(\frac{A}{2}\right)n_{air}+\left(\frac{A}{2}\right)\Delta n\cos\left(kx\right)\right],$$
Here $\lambda_{air}$ is the wavelength of the incident light, the refractive indices of the liquid and air are denoted by $n_{liquid}$ and $n_{air}$, respectively, and we define $\Delta n=\left(n_{liquid}-n_{air}\right)$.

A collimated beam of coherent light that impinges normally (in the *z*-direction) on the corrugated liquid-air interface h(x) in the *x*-*y* plane will be diffracted into the *x*-*z* plane because of the spatially periodic optical path. The relative intensity of the ith diffracted order of monochromatic light transmitted through such a sinusoidal phase transmission grating has an analytical solution and is predicted by the Kirchoff integral to scale as the square of Bessel functions of first kind, $J_{i}^{2}\left(m\right)$, where m is the phase delay excursion in radians [38,39], given from Equation (1) as $m=\pi A\Delta n/\lambda_{air}$. This direct relationship between the argument m of the squares of the Bessel functions that predict the relative intensities of the diffracted orders, and the peak to trough amplitude A of the liquid surface corrugation, provides a facile means to extract the rapidly time varying value of A(t) from optical diffraction measurements.

To explore the dynamics of wrinkle formation and decay we applied an A.C. sinewave voltage with frequency 2.5 kHz to the electrodes to form a film of the liquid TMP-TG-E (volume 0.99 ± 0.05 μL) of thickness $h_{ave}=$ 40 ± 1 μm in the region of the surface that we studied. The voltage was amplitude modulated, using a slower 100 Hz squarewave, alternating between a minimum value $V_{R.M.S.}$ = 46 V (±1 V) and a maximum value of $V_{R.M.S.}$ = 293 V (±1 V). Collimated monochromatic incident light of wavelength $\lambda_{air}$ = 532 nm from a green 10 mW laser diode was transmitted through the liquid and diffracted by the surface wrinkle. With the transitions of the modulation waveforms acting as timing triggers, we used a digital storage oscilloscope to capture the time dependent intensities of the zeroth, 1st and 2nd diffracted orders ($I_{o}\left(t\right)$, $I_{1}\left(t\right)$, and $I_{2}\left(t\right)$, respectively) recorded by three separate photodiodes. The solid line plots in Figure 5a show how the intensities of these three diffracted orders varied with time immediately after the voltage value had increased abruptly from 46 V to 293 V, which corresponds to the dielectrophoresis induced growth of the wrinkle peak to trough amplitude A(t). The solid line plots in Figure 5b show how the intensities of these same three diffracted orders varied with time immediately after the voltage value had decreased abruptly from 293 V to 46 V, corresponding to the decay of the wrinkle peak to trough amplitude A(t) during capillary driven levelling of the surface of the liquid film.

The selected electrical device addressing parameters, including the minimum voltage value at 46 V, the 2.5 kHz A.C. driving voltage that applied the dielectrophoresis forces to the liquid, and the 100 Hz low frequency modulation waveform that provided cycling between alternate 5 ms time periods of growth (293 V) or decay (46 V) of the wrinkle amplitude, were found in combination to deliver on the requirement to maintain constancy of the film thickness *h_ave_* during the measurements. In addition, the selected parameters allow sufficient time for the measurement of growth to saturation and decay to negligible peak to trough amplitude of the surface corrugation during each 5 ms time period. At the 46 V minimum voltage, the equilibrium value of the wrinkle amplitude was sufficiently negligible to enable study of levelling, whilst this value also acted effectively to prevent the liquid film from beginning to de-wet and hence avoided any significant changes in the overall shape of the liquid film during the measurements. The efficacy of this approach was confirmed during the measurements using a CMOS camera (DCC1545M, Thorlabs, Ely, UK) fitted with a 5× objective lens and a 150 mm tube to image the edge of liquid film in the *y*-direction to monitor the overall liquid film shape and the value of $h_{ave}$ at the position on the liquid film at which the collimated incident laser light was transmitted and diffracted. This imaging set-up is shown in Figure 1. 

Figure 6 shows plots of the time dependent amplitude of the periodic wrinkle deformation A(t), during growth towards equilibrium amplitude in Figure 6a, and during decay towards zero in Figure 6b. These curves were obtained by extracting time dependent functions m(t) from the fits shown in Figure 5 of the squares of the Bessel functions $J_{0}^{2}\left(m\left(t\right)\right)$ (shown by open circle symbols), $J_{1}^{2}\left(m\left(t\right)\right)$ (open square symbols), and $J_{2}^{2}\left(m\left(t\right)\right)$ (open diamond symbols), to the separately measured intensities of the diffracted orders, $I_{o}\left(t\right)$, $I_{1}\left(t\right)$, and $I_{2}\left(t\right)$ (shown by solid lines), respectively. As discussed above, the peak to trough wrinkle amplitude A(t) on a film of liquid TMP-TG-E is related to the phase delay excursion *m*(*t*) by the expression $A\left(t\right)=m\left(t\right)\lambda_{air}/\left(\pi \Delta n\right)=$ 3.55 × 10^−7^
*m*(*t*) metres, for transmitted laser light of wavelength $\lambda_{air}$ = 532 nm, with $\Delta n=\left(n_{TMP-TG-E}-n_{air}\right)=$ 0.477, and assuming a spatially periodic wrinkle profile and commensurate optical path variation as given in Equation (1). Taking the measured 0th order diffraction data $I_{o}\left(t\right)$ shown by the open circles in Figure 5a as an example, the solid line through the data shows the fit to the square of the Bessel function of first kind, $J_{0}^{2}\left(m\left(t\right)\right)$, to the normalised data, $I_{o}\left(t\right)$, which yielded the time-dependent function m(t), and hence the plot of A(t) against time t shown by the solid line in Figure 6a. Extracting m(t) from fitting $J_{0}^{2}\left(m\left(t\right)\right)$ to the experimental data is an inverse problem, and the fitting function has an oscillatory dependence on its argument, m(t). We obtained the fit in a straightforward manner by considering each time interval between adjacent maxima and minima in turn.

In Figure 6a the peak to trough amplitude A(t) of a growing wrinkle deformation shows a steep rise from close to zero, A(t=0)≈ 0.1 μm, at t= 0 s, when the voltage was abruptly increased from 46 V to 293 V, with the subsequent rate of increase slowing approaching an equilibrium asymptote of A(t→∞)≈ 3.6 μm. This growth in amplitude is the result of dielectrophoresis forces dynamically creating the wrinkle deformation, under the electrostatic action of the spatially periodic potential applied to the underside of the liquid film, whilst opposed by the Laplace pressure associated with surface tension forces and resisted by the liquid viscosity. Very similar shapes are found, as should be expected, for all three curves, A(t), in Figure 6a that were derived from separate fits to the 0th order diffracted order ($A_{\mathrm{on},i=0}\left(t\right)$, solid line), the 1st order ($A_{\mathrm{on},\;i=1}\left(t\right)$, dashed line), and the 2nd order ($A_{\mathrm{on},i=2}\left(t\right)$, dotted line). The inset graph in Figure 6a compares each of these three curves with the exponential rise to equilibrium function, $A\left(t\right)=A\left(t\to \infty \right)\left[1-e^{-t/\tau_{\mathrm{on}}}\right]$. The linear dependence demonstrates that the exponential rise provides an excellent description of the time dependence of the wrinkle growth. Linear regression analysis gave A(t→∞) = 3.55 μm, 3.61 μm, and 3.62 μm (±0.02 μm) and $\tau_{\mathrm{on}}$ = 3.469 × 10^−4^ s, 3.496 × 10^−4^ s, and 3.456 × 10^−4^ s (±0.002 × 10^−4^ s) for the curves $A_{\mathrm{on},i=0}\left(t\right)$, $A_{\mathrm{on},i=1}\left(t\right)$ and $A_{\mathrm{on},i=2}\left(t\right)$, respectively. The voltage was then abruptly decreased from 293 V to 46 V, and Figure 6b shows the wrinkle amplitude A(t) responding by decaying steeply from initial equilibrium amplitude A(t=0)≈ 3.6 μm, at t= 0 s towards the equilibrium asymptote of A(t→∞)≈ 0.1 μm. Here capillary driven levelling of the surface of the liquid film, driven by the Laplace pressure associated with surface tension forces, is resisted by the liquid viscosity. Figure 6b shows very similar shapes, as should be expected, for the three curves A(t) derived from the separate fits to the 0th order diffracted order ($A_{\mathrm{off},i=0}\left(t\right)$, solid line), the 1st order ($A_{\mathrm{off},\;i=1}\left(t\right)$, dashed line), and the 2nd order ($A_{\mathrm{off},i=2}\left(t\right)$, dotted line). The inset graph in Figure 6b compares these three curves with the exponential decay function,$\;A\left(t\right)=A\left(t\to \infty \right)e^{-t/\tau_{\mathrm{off}}}$. The linear dependence demonstrates that the exponential decay provides an excellent description of the time dependence of the wrinkle levelling. Linear regression analysis gave $\tau_{\mathrm{off}}$ = 4.410 × 10^−4^ s, 4.438 × 10^−4^ s, and 4.394 × 10^−4^ s (±0.003 × 10^−4^ s) for $A_{\mathrm{off},i=0}\left(t\right)$,$\;A_{\mathrm{off},i=1}\left(t\right)$ and $A_{\mathrm{off},i=2}\left(t\right)$, respectively.

Figure 7 plots the values of the time constants for the dielectrophoresis induced formation and growth in the amplitude of the wrinkle deformation, $\tau_{\mathrm{on}}$ (filled diamonds), and the values of the time constants for the capillary force driven decay in the wrinkle amplitude, $\tau_{\mathrm{off}}$ (closed diamonds), for a periodic wrinkle deformation of pitch p = 160 μm on different TMP-TG-E liquid films having thicknesses in the range from $h_{ave}\;$= 32 ± 1 µm up to $h_{ave}\;$= 70 ± 1 µm. These time constant values were obtained from exponential rise to equilibrium or exponential decay function fits to time dependent peak to trough wrinkle amplitude A(t) curves obtained from the 0th order diffraction data, using the experimental parameters and the data analysis and fitting processes as described above and as depicted in Figure 5 and Figure 6.

We first consider the time constant, $\tau_{\mathrm{off}}$, for the exponential decay of the amplitude of the liquid surface wrinkle. The capillary force driven levelling of a sinusoidal distortion on a clean surface of a thin liquid layer was predicted by Orchard [11] to be described by an exponential decay with characteristic decay time, $\tau_{\mathrm{off}}$, given by the analytical expression in Equation (2).
(2)$$\tau_{\mathrm{off}}\left(kh_{ave}\right)=\frac{2\eta }{\gamma kf\left(kh_{ave}\right)},\;f\left(kh_{ave}\right)=\frac{\tanh\left(kh_{ave}\right)-\left(kh_{ave}\right){\left[\sec h\left(kh_{ave}\right)\right]}^{2}}{1+{\left(kh_{ave}\right)}^{2}{\left[\sec h\left(kh_{ave}\right)\right]}^{2}},$$

The function tends to the limit $f\left(kh_{ave}\right)\to 2{\left(kh_{ave}\right)}^{3}/3$ when $\left(kh_{ave}\right)\ll 1$, so that when the pitch of the wrinkle is much greater than the thickness of the liquid film, $p\gg h_{ave}$ (where k=2π/p), it simplifies to the familiar approximate expression $\tau_{\mathrm{off}}=3p^{4}\eta /\left(16\gamma \pi^{4}h_{ave}^{3}\right)$. The solid line on Figure 7 shows a plot of the full Equation (2) using the physical parameters for our experimental system (see experimental section) as a function of the film thickness, $h_{ave}$. There is close agreement between the solid line plot of Equation (2), and the experimental decay time constant data, shown by the open diamonds, across the range of film thicknesses explored. This agreement also extends to the lowest film thickness values where the initial shape of the wrinkle on the film deviates more significantly from being sinusoidal, as illustrated by the results of the OCT study of equilibrium surface profile shapes on a thin film as depicted earlier in Figure 4b. The levelling that we have studied was initiated starting from wrinkle amplitudes and profiles that had been created using dielectrophoresis forces. The close agreement between the established theory for levelling with the experimental results gives confidence that potential sources of systematic error have been avoided in our experiments. For example, there is negligible ohmic heating by the electrodes since this would have been evident by a reduction in the liquid viscosity and hence the time constant $\tau_{\mathrm{off}}$. 

We now consider the time constant,$\;\tau_{\mathrm{on}}$, for the dielectrophoresis driven increase in the amplitude of the liquid surface wrinkle. We use the theoretical approach to the formation of a wrinkle profile at the interface of two dielectric fluids by dielectrophoresis forces is described by Chappell and O’Dea [33]. This approach uses Nyström based boundary integral methods to solve the equations of Stokes flow coupled with an approximate electrostatic potential model to obtain the potential on the fluid-air interface. This permits numerical solutions to the time dependent formation and decay of wrinkle profiles when supplemented with a kinematic boundary condition to model the interface motion resulting from the fluid velocity field. This approach uses a sinusoidal spatially periodic potential and sinusoidal wrinkle profile [37]. We first checked that the numerical model gave the same results as predicted by Equation (2) for the exponential decay of the amplitude of a liquid surface wrinkle, $\tau_{\mathrm{off}}\left(h_{ave}\right)$, and this gave an exact match to the solid line in Figure 7. We then performed numerical modelling to find the timescales for the increase in a sinusoidal wrinkle amplitude, $\tau_{\mathrm{on}}\left(h_{ave}\right)$, resulting in the dashed line in Figure 7. Both the experimental data and the model follow the same trend of decreasing time constant for increasing film thickness, with the growth time constant being quicker than the decay time constant for all film thickness studied, i.e., $\tau_{\mathrm{on}}<\tau_{\mathrm{off}}$. The difference between these time constants can be understood by consideration of the strength of the forces which drive the formation and decay of the wrinkle. Prior to switching the applied voltage to the “on” state the liquid-air interface is predominantly flattened by surface tension. Once the device is switched to the “on” state a Maxwell stress [40] is induced at the interface which is resisted by the Laplace pressure. For the interface to distort, the Maxwell stress must overcome the Laplace pressure, therefore the force which forms the film is stronger than the force which drives the decay of the film. As the thickness of the liquid film decreases, the liquid-air interface becomes closer to the decaying electric field and this increases the strength of the Maxwell stress at the interface, as well as increasing the amplitude of the wrinkle formed since the maximum voltage was kept constant at 293 V for the full range of film thicknesses. Therefore, the difference between $\tau_{\mathrm{on}}$ and $\tau_{\mathrm{off}}$ is expected to increase with decreasing film thickness, as observed in the experiments. However, as the thickness of the film decreases, viscous dissipation within the film increases resulting in a retarding of both the formation and decay time.

As well as the numerical model of dielectrophoresis driven wrinkling correctly predicting that $\tau_{\mathrm{on}}<\tau_{\mathrm{off}}$, we find excellent agreement between the experimental data and the numerical modelling prediction for the value of the growth time constant $\tau_{\mathrm{on}}$ for film thicknesses above $h_{ave}>$ 50 μm. For liquid films with thickness below 50 μm we find that the model predicts a higher growth time constant $\tau_{\mathrm{on}}$ compared to the experimentally measured switch on time. We found that the inclusion of higher harmonics in the potential and surface shape profile in the numerical simulation model led to a still further increase in the value of the predicted growth time constant, and hence did not account for this difference. 

## 4. Discussion

We have demonstrated that Optical Coherence Tomography (OCT) imaging can be used to make precise measurements of a static wrinkle profile produced by dielectrophoresis forces on the surface of a thin film of the isotropic liquid TMP-TG-E (trimethylolpropane triglycidyl ether) in equilibrium. OCT techniques have yielded the profiles of micrometre scale features on a liquid-air surface, with sub-micrometer axial precision. Periodic wrinkle profiles for this type of experimental system (but using different liquids) have previously been obtained using a Mach–Zehnder interferometer [16] using collimated monochromatic light transmitted through the liquid film. The Mach–Zehnder technique works in transmission and suffers from loss of accuracy for high surface deformation because of the optical focusing properties of the wrinkles which act as lenticular array of cylindrical micro-lenses. Moreover, extracting liquid-air surface profiles using interferometry techniques in transmission is generally only possible for transparent non-scattering isotropic liquid films, of the type that have been the subject of our study in this report. Key additional advantages of OCT are that it can distinguish multiple optical interfaces in a system and could equally be applied to scattering or turbid liquids, or to complex liquids including anisotropic birefringent liquids such as liquid crystals in which there are sharp and localized spatial variations of the local optical axis within the liquid film [41] that would make transmission-based measurements challenging to interpret.

Our static OCT imaging study provided detailed knowledge of the wrinkle shapes and amplitudes which informed the choice of the driving voltage and the range of TMP-TG-E liquid film thicknesses (from $h_{ave}\;$= 32 µm up to 70 µm) that we used in our further study of the sub-millisecond dynamics of wrinkle formation and decay. We developed and employed facile time-resolved diffraction, and associated data analysis and techniques which yielded the time dependent amplitude A(t) of a dielectrophoresis induced periodic wrinkle during its growth towards equilibrium immediately after a voltage was abruptly applied, and also during its decay after a voltage was abruptly reduced. Both growth and decay of the amplitude of the periodic wrinkle were found to be exponential in nature with characteristic timescales, $\tau_{\mathrm{on}}\;$ and $\tau_{\mathrm{off}}$, respectively, with decreasing values as the film thickness increased, and with $\tau_{\mathrm{on}}<\tau_{\mathrm{off}}$ for all film thickness studied. This observation was reproduced by our numerical modelling, following *Chappell* and *O’Dea* [33], using Nyström based boundary integral methods to solve the equations of Stokes flow. This theoretical approach reproduced the measured growth time constant $\tau_{\mathrm{on}}$ for film thickness values above $h_{ave}>$ 50 μm but over-predicted the growth rate for liquid films where $h_{ave}<$ 50 μm. For decay, we found excellent quantitative agreement with the theoretical predictions of capillary driven levelling of *Orchard* [11].

## Figures and Tables

**Figure 1 micromachines-12-01583-f001:**
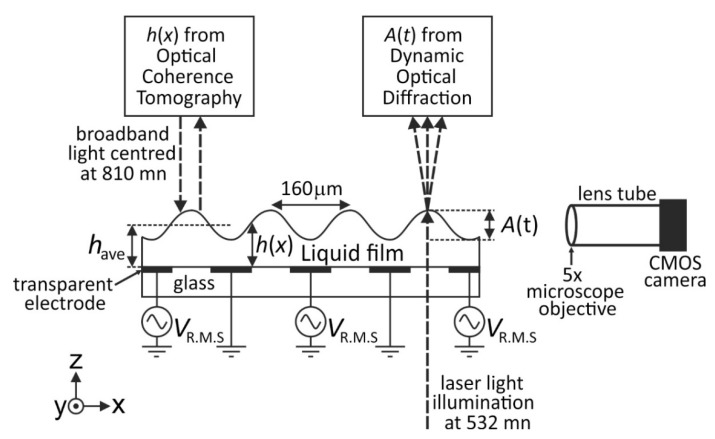
Schematic diagram of the experimental system and the device substrate geometry. The Optical Coherence Tomography (OCT) apparatus is shown in detail in the schematic diagram in subsequent Figure 2. The dynamic optical diffraction measurements with auxiliary side-view CMOS camera imaging were performed separately to the OCT measurements.

**Figure 2 micromachines-12-01583-f002:**
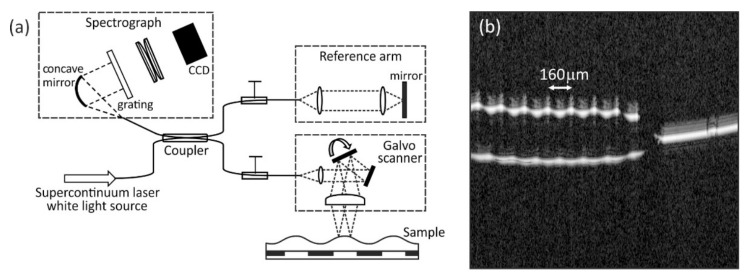
(**a**) Schematic diagram of the Optical Coherence Tomography (OCT) imaging apparatus that was used to measure the equilibrium surface height profile *h*(*x*) of thin films of the liquid TMP-TG-E on the device substrate; (**b**) A raw “BScan” image of the TMP-TG-E film obtained using the OCT apparatus with a constant static A.C. voltage $V_{R.M.S.}$ = 250 V (1 kHz) applied to the electrodes of the device.

**Figure 3 micromachines-12-01583-f003:**
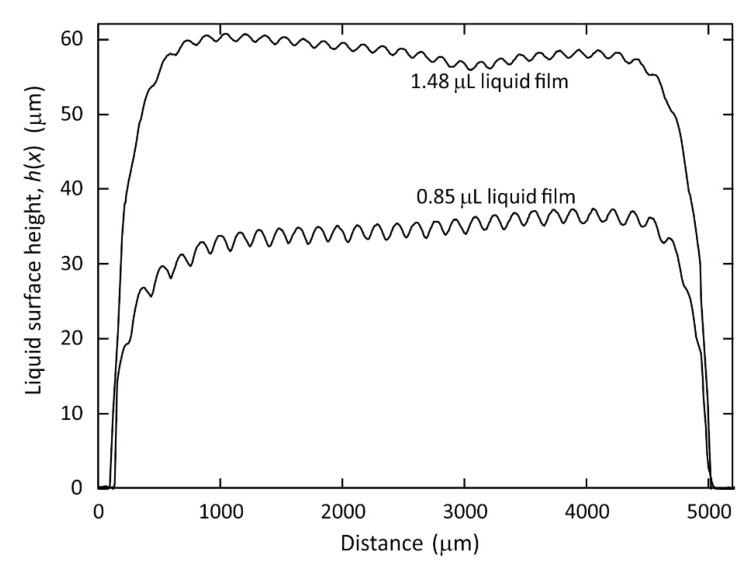
Plot of the static equilibrium surface height profile *h*(*x*) of two different spread thin films of the liquid TMP-TG-E, measured using Optical Coherence Tomography imaging. Two different volumes of liquid were dispensed onto the substrate, 1.48 ± 0.05 μL (upper plot) and 0.85 ± 0.05 μL (lower plot). In each case, a constant A.C. voltage $V_{R.M.S.}$ = 200 V (1 kHz) applied to the electrodes resulted in a spread liquid film that exhibited a static periodic surface wrinkle deformation.

**Figure 4 micromachines-12-01583-f004:**
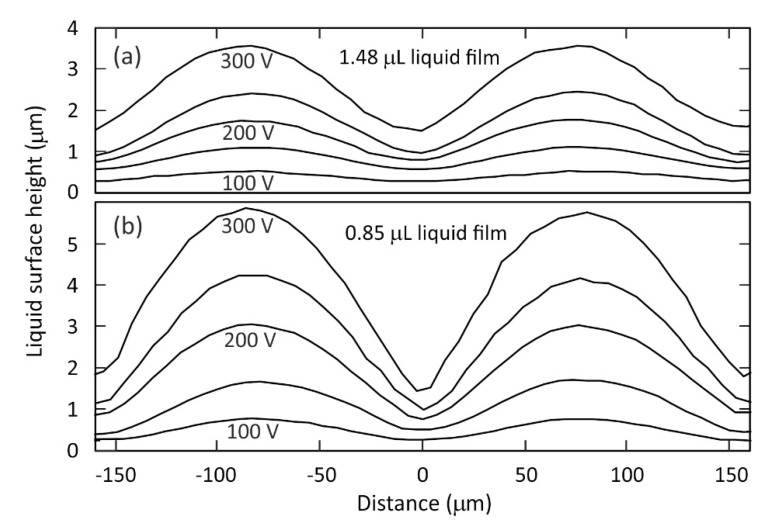
Plot of the static equilibrium surface height profile *h*(*x*) of two different spread thin films of the liquid TMP-TG-E measured using optical coherence tomography imaging, zoomed in from Figure 3. Profiles are shown on films formed from two different dispensed volumes of liquid: (**a**) 1.48 ± 0.05 μL, and (**b**) 0.85 ± 0.05 μL. The equilibrium periodic wrinkle deformation profiles are shown for 5 different voltages (R.M.S. 1 kHz) applied to the electrodes under the liquid, $V_{R.M.S.}$ = 100, 150, 200, 250, 300 V. The average vertical positions of the profiles are offset for clarity.

**Figure 5 micromachines-12-01583-f005:**
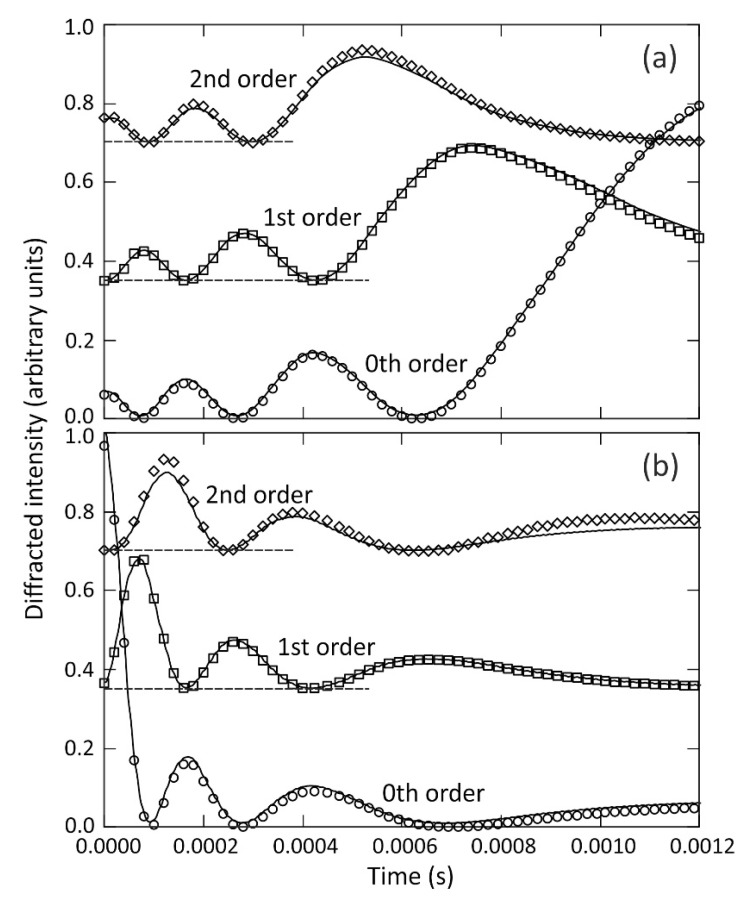
The solid lines show the measured time dependent intensity of laser light at 532 nm diffracted into the 0th, 1st and 2nd orders, by transmission through a dielectrophoresis induced periodic wrinkle deformation on a TMP-TG-E liquid film of thickness $h_{ave}$ = 40 ± 1 µm. The 1st and 2nd order plots use the same vertical scale as the 0th order, but are shifted upwards by 0.35 and 0.70 units, respectively, for clarity. In plot (**a**) the 2.5 kHz A.C. sinewave driving voltage had been increased abruptly from 46 V to 293 V at time t=0, and in plot (**b**) the voltage value had been decreased abruptly from 293 V to 46 V at time t=0. The open symbols show the fits to squares of Bessel functions of first kind, $J_{i}^{2}\left(m\right)$, with i=0 (circles), i=1 (squares), and i=2 (diamonds).

**Figure 6 micromachines-12-01583-f006:**
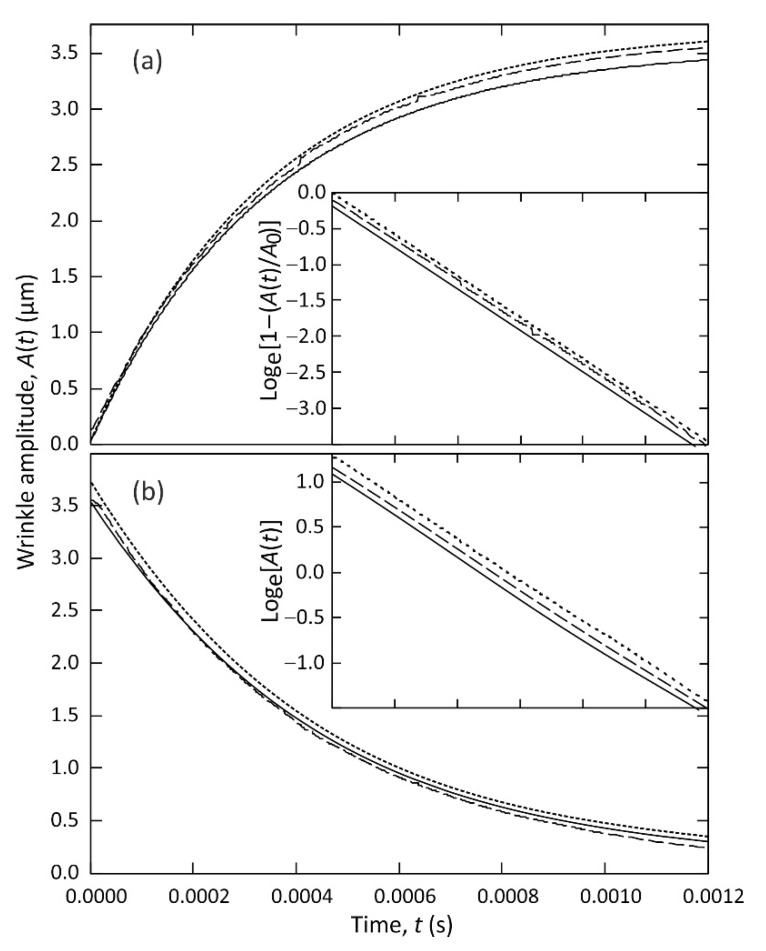
The measured time dependence of the peak to trough amplitude A(t) of a growing (**a**) and decaying (**b**) periodic wrinkle deformation on a TMP-TG-E liquid film of thickness $h_{ave}$ = 40 ± 1 µm. A(t) was extracted from fitting to the optical diffraction data shown in Figure 5, with plot (**a**) derived from the fits shown in Figure 5a, and plot (**b**) derived from the fits shown in Figure 5b, where the solid line shows the fit to the 0th order, the dashed line the 1st order, and the dotted line the 2nd order. The inset graphs show fits of the A(t) results (same time axis) to, (**a**) an exponential rise to equilibrium function, and (**b**) to an exponential decay function, with A(t) functions from the 1st and 2nd orders displaced vertically by 0.1 and 0.2 units, respectively, for clarity.

**Figure 7 micromachines-12-01583-f007:**
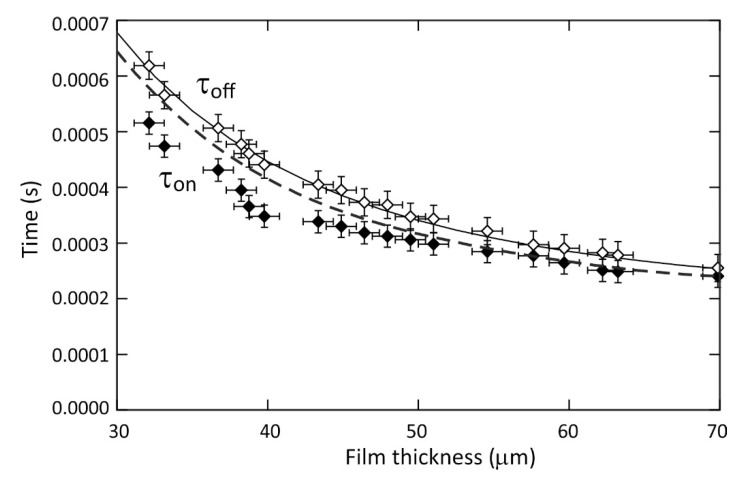
Measured time constant values for the exponential rise of the amplitude A(t) of a liquid surface wrinkle towards equilibrium, $\tau_{\mathrm{on}}$ (filled diamonds), and for the exponential decay of the wrinkle amplitude, $\tau_{\mathrm{off}}$ (open diamonds), for periodic wrinkle deformations of pitch p = 160 μm on different thickness ($h_{ave}$) TMP-TG-E liquid films. The solid line shows the theoretical prediction by Orchard [11], $\tau_{\mathrm{off}}\left(kh_{ave}\right)$, for the surface tension driven levelling of a liquid surface corrugation, where k=2π/p is a constant. The dashed line shows our theoretical prediction of the wrinkle growth time constant, $\tau_{\mathrm{on}}\left(kh_{ave}\right)$, from numerical solution of the equations of Stokes flow with electrostatic forcing.

## Data Availability

The data presented in this study are available on request from the corresponding author.

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
