# Peer review of "Static and Dynamic Optical Analysis of Micro Wrinkle Formation on a Liquid Surface"

_micromachines, 2021, doi:10.3390/mi12121583_

Round 1

Reviewer 1 Report

In this work the authors study the wrinkle formation of a liquid film under dielectrophoretic forces. The authors make use of Optical Coherence Tomography (OCT) to  study both statically and dynamically the wrinkles. The authors find that the amplitude growth and decay of the wrinkles is exponential and depends of the wrinkles depends on the thickness of the liquid films. The findings are of interest to the community owing to its importance from both fundamental and application viewpoints. The experiments are sound; the analytical model captures the observations with reasonable fidelity. However, I have some comments which might improve the understanding of the paper.

  • Why was that specific range of frequencies chosen for the study? Is there any influence of the frequency on the wrinkles properties (width, growth rate, etc.)
  • The authors mentioned “indicating that when we subsequently remove or significantly reduce the voltage, the levelling of these induced surface wrinkle distortions will be dominated by capillary forces”. Similar claims are stated on the paragraph starting in line 344. Can the authors comment on the effect of wettability of the substrate on the wrinkle formation and decay?
  • In the discussion section the sentence starting in line 468 and ending in line 472, the authors state that OCT can be applied to scattering and turbid liquids. However, I do not see how is this related to the study presented here. Is the liquid used in here is turbid or scattering?
  • In line 129 a range of used voltage would be helpful to the reader.
  • The authors mention “Higher voltages then resulted in the formation of a clear wrinkle” What is the threshold voltage for this wrinkle formation?
  • In lines 196-198 the authors say that the amplitude of the wrinkles at equilibrium are larger for the liquid film that has a lower average thickness. It would be good if this difference is quantified and stated (maybe in terms of percentages).
  • In lines 180 and 181 the authors say “The liquid rapidly spreads...” What is the order of magnitude of the time the liquid takes to spread?
  • A diagram of the experimental setup and a typical image obtained from OCT would be helpful to reader.
  • An figure showing the process that you describe on the lines 310-315 would be helpful.
  • The sentence that starts in line 183 and ends in line 186 starting with “We analyzed...” is confusing.
  • Image quality (dpi) of the figures needs to be improved.

Author Response

Author response

Thank you to the reviewer for their careful and through reading of our manuscript. We have made amendments with the aim of fully addressing all of their comments and suggestions.

  • Reviewer #1 comment 1: Why was that specific range of frequencies chosen for the study? Is there any influence of the frequency on the wrinkles properties (width, growth rate, etc.)

Dielectrophoresis forces are independent of polarity and so, in theory, either D.C. or A.C. voltages can be used for the dielectrophoresis actuation of liquids and to create voltage forced wrinkling. Hence, in theory there should be no influence of the frequency on the wrinkle properties, which show a root mean square amplitude response in the case of A.C. voltages. However, in practice A.C. voltages are often used in order to avoid effects due to the field induced migration of free charges that are inevitably present in the liquid. Low frequencies, typically below 100 Hz, are avoided to reduce charge migration. High frequencies, typically above 100 kHz, are also avoided due to the finite slew rate of the amplifier and to avoid voltage losses due to the finite conductivity of the indium tin oxide electrodes. Furthermore, for our detailed study of the wrinkle properties the frequency of the voltage should be sufficiently high to prevent significant change in wrinkle amplitude during each half period. The A.C. driving voltage frequency should not be significantly lower than the reciprocal of the wrinkle formation and decay time constants.  

Amendment made: In paragraph 3 of section 2 we have added: “Dielectrophoresis forces are independent of polarity and so, in theory, either D.C. or A.C. voltages can be used create the wrinkle deformation. However, in practice A.C. voltages are used to the avoid the D.C. and low frequency shielding effects of the field induced migration of free charges that are inevitably present even in the low electrically conductivity liquid TPM-TG-E. We used voltages of 1 kHz (section 3.1) and 2.5 kHz (section 3.2) which provides a suitable compromise that avoids low frequencies that would have caused charge migration and also would have significantly modulated the wrinkle amplitude during each half period, and that also avoids high frequencies where the finite slew rate of the amplifier and the finite conductivity of the indium tin oxide electrodes would have caused signal losses.”   

  • Reviewer #1 comment 2: The authors mentioned “indicating that when we subsequently remove or significantly reduce the voltage, the levelling of these induced surface wrinkle distortions will be dominated by capillary forces”. Similar claims are stated on the paragraph starting in line 344. Can the authors comment on the effect of wettability of the substrate on the wrinkle formation and decay?

Thank you for this comment. We have taken careful steps to avoid film dewetting, which is a slow process (of order 1 second, please see the response below to comment 7), which is why the minimum voltage used to study the wrinkle deformation is 100 V in section 3.1, and is 46 V (which is only applied for half of the 100 Hz modulation period) in section 3.2. We can now see that mentioning “removing the voltage” completely in our manuscript is confusing and that we need to make this clearer.

Amendment made: In paragraph 5 of section 1 we have changed “… after the voltage is removed, provides … ” to “… after the voltage is significantly reduced (we do not completely remove the voltage to avoid film dewetting), provides …”

Amendment made: In paragraph 3 of section 2 we have changed “… when we subsequently remove or significantly reduce the voltage … ” to “… when we subsequently significantly reduce the voltage …”

  • Reviewer #1 comment 3: In the discussion section the sentence starting in line 468 and ending in line 472, the authors state that OCT can be applied to scattering and turbid liquids. However, I do not see how is this related to the study presented here. Is the liquid used in here is turbid or scattering?

We agree with the reviewer that this statement is confusing, and so we have modified this sentence.

Amendment made: In paragraph 1 of section 4 we have changed “Moreover, extracting liquid-air surface profiles using interferometry techniques in transmission is only possible for transparent non-scattering isotropic liquid films. Key advantages of OCT are that it can distinguish multiple optical interfaces in a system and can equally be applied to scattering or turbid liquids, and to complex liquids …” to “Moreover, extracting liquid-air surface profiles using interferometry techniques in transmission is only possible for transparent non-scattering isotropic liquid films, of the type that have been the subject of our study in this report. Key additional advantages of OCT are that it can distinguish multiple optical interfaces in a system and could equally be applied to scattering or turbid liquids, or to complex…” 

  • Reviewer #1 comment 4: In line 129 a range of used voltage would be helpful to the reader.

We agree with the reviewer.

Amendment made: In paragraph 3 of section 2 we have changed “The voltage waveform (R.M.S. voltage …” to “The voltage waveform up to 300 V (R.M.S. voltage…” 

  • Reviewer #1 comment 5: The authors mention “Higher voltages then resulted in the formation of a clear wrinkle” What is the threshold voltage for this wrinkle formation?

There is no threshold voltage for wrinkle formation on a thin spread film of constant thickness. Our statement here in cognizant of the fact that at low voltages, below 50 V, the liquid film would begin to dewet (see our answer to comment 2 above) so we started at this voltage first.

Amendment made: In paragraph 3 of section 2 we have changed “Higher voltages then resulted in the formation of a clear “wrinkle” deformation …” to “The applied voltage resulted in the formation of a clear “wrinkle” deformation …”  

  • Reviewer #1 comment 6: In lines 196-198 the authors say that the amplitude of the wrinkles at equilibrium are larger for the liquid film that has a lower average thickness. It would be good if this difference is quantified and stated (maybe in terms of percentages).

We agree with the reviewer.

Amendment made: In paragraph 2 of section 3.1 we have added “For example, with an applied voltage of  = 200 V the peak to trough wrinkle amplitude is  = 2.25 ± 0.05 mm on the film with thickness 30 μm to 34 μm in Figure 4(b), compared with  = 0.93 ± 0.05 mm on the film with thickness 57 μm to 60 μm in Figure 4(a).”  

  • Reviewer #1 comment 7: In lines 180 and 181 the authors say “The liquid rapidly spreads...” What is the order of magnitude of the time the liquid takes to spread?

For the liquid used TMP-TG-E (trimethylolpropane triglycidyl ether) spreading occurs within 1 second of the voltage being applied. Voltage forced wetting is a significantly slower process compared to voltage forced wrinkling, because wetting involves contact line movement whereas wrinkling does not.

Amendment made: In paragraph 1 of section 3.1 we have changed “The liquid rapidly spread into a thin film and developed” to “The liquid rapidly spread, within 1 second of the voltage being applied, into a thin film and developed”.

  • Reviewer #1 comment 8: A diagram of the experimental setup and a typical image obtained from OCT would be helpful to reader.

We agree with the reviewer.

Amendment made: We have provided a brand new diagram, Figure 2 in the revised manuscript. This provides a schematic diagram of the Optical Coherence Tomography apparatus in Figure 2(a), plus an example B-scan “raw image” of the device with a wrinkled surface in Figure 2(b).

Amendment made: To aid the clarity of the manuscript we have also now given the schematic diagram of the device and of the electrical addressing geometry separately in a revised Figure 1. In the previous version of the manuscript this schematic diagram was inset into the old Figure 1. Our revised Figure 1 now indicates the Optical Coherence Tomography measurement of h(x) and the Dynamic Optical Diffraction measurement of A(t), as well as the auxiliary side camera used alongside the diffraction measurement.

  • Reviewer #1 comment 9: An figure showing the process that you describe on the lines 310-315 would be helpful.

We are happy to comply with this helpful suggestion.

Amendment made: We have shown the side view monitoring camera in our revised Figure 1, as discussed above this now shows the auxiliary side camera used alongside the diffraction measurement.

  • Reviewer #1 comment 10: The sentence that starts in line 183 and ends in line 186 starting with “We analyzed...” is confusing.

We agree with the reviewer.

Amendment made: In paragraph 1 of section 3.1 we have changed “We analyzed the OCT cross-section image of the liquid film and fitted the peak corresponding to the air-liquid interface in each of depth profile and the peak position yielded the equilibrium surface height profile of the film shown as the upper labelled plot on the graph of h(x) versus distance x in Figure 3” to “We analyzed the OCT cross-section BScan image of the liquid film. Firstly, we corrected the curvature of the field as shown on the right hand side of the example BScan profile shown Figure 2(b). We then fitted the depth profile (AScan) for the peaks corresponding to the air-liquid and air-substrate interfaces at each x position to subpixel accuracy. After accounting for the refractive index of the liquid TMP-TG-E this gave the film thickness at each x position. Repeating this procedure for a range of x positions yielded the equilib-rium surface height profile of the film shown as the upper labelled plot on the graph of h(x) versus distance x in Figure 3.”

  • Reviewer #1 comment 11: Image quality (dpi) of the figures needs to be improved.

We are happy to comply with this helpful suggestion.

Amendment made: We have increased the resolution quality of all of our jpeg figures that are embedded into the revised manuscript.

Reviewer 2 Report

Please see the attach PDF for the comments.

Author Response

Author response

Thank you to the reviewer for their careful and through reading of our manuscript. We have made amendments with the aim of fully addressing all of their comments and suggestions.

Reviewer #2 Query 1: Figure 2 in the manuscript, why initial thinner liquid leads to the deep wrinkle (higher amplitude) whereas thick film leads to the shallow wrinkle?

We agree with the reviewer that this point is worthy of clarification.

Amendment made: In paragraph 2 of section 3.1 we have added “For example, with an applied voltage of  = 200 V the peak to trough wrinkle amplitude is  = 2.25 ± 0.05 mm on the film with thickness 30 μm to 34 μm in Figure 4(b), compared with  = 0.93 ± 0.05 mm on the film with thickness 57 μm to 60 μm in Figure 4(a).”  

Amendment made: In paragraph 2 of section 3.1 we have added “This effect is a consequence of the fact that the electric fields above the electrodes decay exponentially with distance in the z direction. Hence for a thinner liquid film the air-liquid interface is closer to the electrodes compared to a thicker film, and so the former experiences a substantially stronger dielectrophoresis force to create the larger wrinkle deformation.”  

Reviewer #2 Suggestion : Cite some recent review articles on dielectrophoresis in the introduction section such as Barman et al. Micromachines 2019, 10, 329

We agree with the reviewer.

Amendment made: We have added the following reference:

Barman, J.; Shao, W.; Tang, B.; Yuan, D.; Groenewold, J.; Zhou, G.; Wettability Manipulation by Interface-Localized Liquid Dielectrophoresis: Fundamentals and Applications. Micromachines 2019, 10(5), 329. [https://doi.org/10.3390/mi10050329]